# *Decoupling Knowledge from Memorization:* Retrieval-augmented Prompt Learning

**Xiang Chen**[1,2]*, **Lei Li**[1,2]*, **Ningyu Zhang**[1,2]†, **Xiaozhuan Liang**[1,2], **Shumin Deng**[1,2],
**Chuanqi Tan**[3], **Fei Huang**[3], **Luo Si**[3], **Huajun Chen**[1,2]†
[1]Zhejiang University & AZFT Joint Lab for Knowledge Engine, China
[2]Hangzhou Innovation Center, Zhejiang University, China
[3]Alibaba Group, China
{xiang_chen, leili21, zhangningyu, liangxiaozhuan, 231sm, huajunsir}@zju.edu.cn,
{chuanqi.tcq, f.huang, luo.si}@alibaba-inc.com

## Abstract

Prompt learning approaches have made waves in natural language processing by inducing better few-shot performance while they still follow a parametric-based learning paradigm; the oblivion and rote memorization problems in learning may encounter unstable generalization issues. Specifically, vanilla prompt learning may struggle to utilize atypical instances by rote during fully-supervised training or overfit shallow patterns with low-shot data. To alleviate such limitations, we develop RETROPROMPT with the motivation of decoupling knowledge from memorization to help the model strike a balance between generalization and memorization. In contrast with vanilla prompt learning, RETROPROMPT constructs an open-book knowledge-store from training instances and implements a retrieval mechanism during the process of input, training and inference, thus equipping the model with the ability to retrieve related contexts from the training corpus as cues for enhancement. Extensive experiments demonstrate that RETROPROMPT can obtain better performance in both few-shot and zero-shot settings. Besides, we further illustrate that our proposed RETROPROMPT can yield better generalization abilities with new datasets. Detailed analysis of memorization indeed reveals RETROPROMPT can reduce the reliance of language models on memorization; thus, improving generalization for downstream tasks[3].

## 1 Introduction

Large parametric language models [45, 7, 22, 31] have achieved dramatic empirical success in natural language processing (NLP). Notably, pre-trained language models (PLMs) have learned a substantial amount of in-depth knowledge from data, and have archived tremendous promise in few-shot/zero-shot learning ability with the natural language prompts [12, 50, 57]. However, Recent studies [37, 39, 59] observe that prompt learning with PLMs usually generalizes unstably in an extremely low-resource setting or emerging domains. One potential reason is that, it is non-trivial for parametric models to *learn rare or hard patterns well with rote memorization*, thus, resulting in inefficient generalizable performance.

Intuitively, if we regard the whole training set as a *book* and the test phase as the *examination*, the current training-test procedure of prompt learning (based on batch data training) can be viewed as

---

* Equal contribution.
† Corresponding Author.
[3]Code is available in `https://github.com/zjunlp/PromptKG/tree/main/research/RetroPrompt`.

*page-by-page memorization* and *closed-book examination* [42]. During training, vanilla prompt learning may struggle to memorize atypical instances in a fully-supervised setting or overfit shallow patterns with low-shot data [62, 9]. Specifically, recent studies[10, 11] have proposed a long-tail theory, which notes that when the training set has a long-tail distribution and contain small "sub-populations" with atypical instances, then PLMs indeed predict on the test data through rote memorizing these atypical instances rather than learning the common patterns [62, 56].

The limitations of rote memorization remind us of the human learning process of *"learn by analogy"* and the proverb that *"the palest ink is better than the best memory"*. Note that humans can perform associative learning to recall relevant skills in deep memories for reinforcing each other, thus, owning the extraordinary abilities to solve few-shot and zero-shot tasks. Motivated by these, we endeavor to improve the generalization ability of prompt learning with retrieval and association. Our intuition is that the difficulty of resolving the above limitations can be substantially alleviated if we can decouple the knowledge from memorization by constructing *an open-book knowledge-store* from the training data; thus, referring to related knowledge could provide a strong enhancement signal to help the model strike a balance between generalization and memorization.

Specifically, we introduce a novel retrieval-augmented framework based on prompt learning (**RETROPROMPT**) as shown in Figure 1. The open-book knowledge store $(\mathcal{K}, \mathcal{V})$, defined as the set of *key: prompt-based example embeddings* and *value: corresponding label words* constructed from the training data, are served as additional references for the model to decouple knowledge from pure memorization to some extent. Specifically, to integrate retrieved knowledge into the input, **Firstly**, we de-

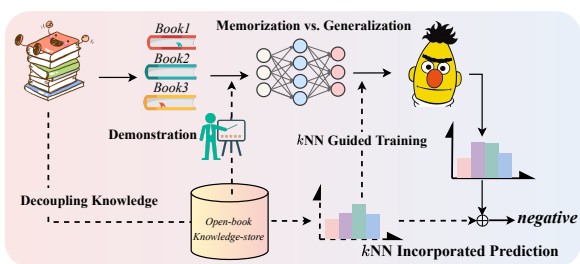

Figure 1: Decoupling knowledge from memorization.

sign to incorporate neural demonstrations into the input sequences as in-context augmentation, where the demonstration is retrieved from the knowledge-store. **Then**, we apply a non-parametric algorithm $k$NN over the input query and knowledge store, and regard $k$NN results as an indication of easy vs. hard instances. Moreover, we automatically force the model to emphasize the hard instances identified by $k$NN by assigning a scaling during training. **Lastly**, the $k$NN results are further employed at the output of the PLM head to participate in masked prediction. The model conducts inference through linearly interpolating the non-parametric nearest neighbor distribution with the output of prompt learning, which regards the Top-$k$ nearest reference instances as cues from $(\mathcal{K}, \mathcal{V})$.

The considerable performance gains on nine tasks in few-shot and zero-shot settings demonstrate that our systemic retrieval mechanism helps the model generalize better with scarce data. Experiments in the fully-supervised setting with long-tail distribution illustrate that our RETROPROMPT can deal with atypical instances more robustly. We further adopt self-influence [27] as our memorization scoring function to analyze the memorization process between fine-tuning, prompt learning and our RETROPROMPT. The final analysis results show that 1) the training samples having the highest memorization scores are mostly atypical, 2) RETROPROMPT generalize better than fine-tuning and convention prompt-tuning with decoupling knowledge from memorization to alleviate the rote of PLMs. In a nutshell, our work may open up new avenues to improve the generalization of prompting PLMs by decoupling knowledge from memorization.

## 2 Preliminaries of Prompt Learning

Assuming that $\mathcal{M}$, $\mathcal{T}$ respectively denotes the PLM and the template function for prompt tuning. Formally, the text classification task takes a query sentence $\boldsymbol{x} = (x_0, x_1, ..., x_n)$ as input. Then, classify it into the label $y \in \mathcal{Y}$. While prompt learning converts the task into a MLM problem with *cloze-style* objectives. Specifically, the template function $\mathcal{T}$ inserts text pieces into $\boldsymbol{x}$ as $\hat{\boldsymbol{x}} = \mathcal{T}(\boldsymbol{x})$, where $\hat{\boldsymbol{x}}$ refers to the input of $\mathcal{M}$ with a [MASK] token. For instances, when we have to classify the text $\boldsymbol{x} =$"The movie makes absolutely no sense." into label NEGATIVE (labeled as 0) or POSITIVE (labeled as 1), we wrap it into

$$\hat{\boldsymbol{x}} = [\text{CLS}]\, \boldsymbol{x} \text{ It was } [\text{MASK}]\,[\text{SEP}] \qquad (1)$$

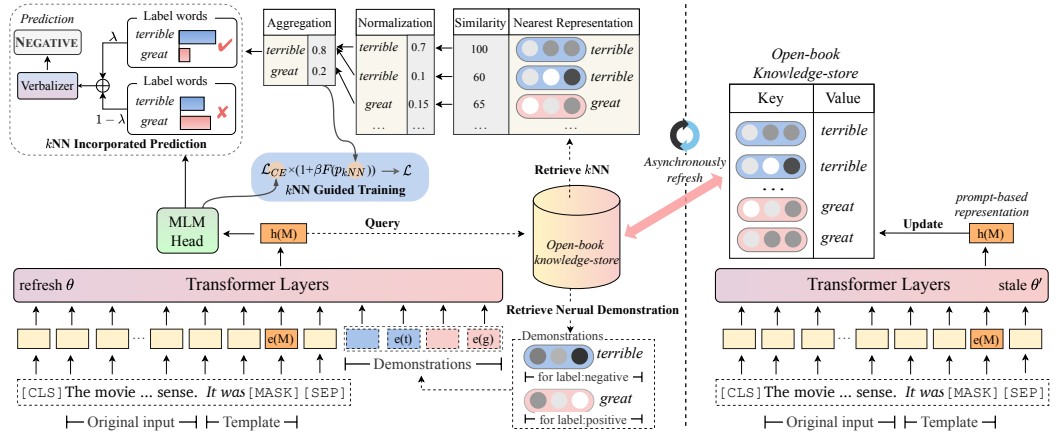

a. Retrieval-augmented prompt learning      b. Creation and refresh of open-book knowledge-store

Figure 2: Overview of RETROPROMPT. Note that $e(\cdot)$ denotes word embedding function in the PLM $\mathcal{M}$, while "M","t" and "g" in $e(\cdot)$ specifically refers to "[MASK]", "terrible" and "great".

The verbalizer $f\colon \mathcal{Y} \mapsto \mathcal{V}$ is defined as a mapping from the label space $\mathcal{Y}$ to those words in the vocabulary, which constructs the *label word* set $\mathcal{V}$. The base component of $\mathcal{M}$ produces the sequence representation over $\hat{\boldsymbol{x}}$, and we choose the hidden vector at the [MASK] position as the contextual representation $\boldsymbol{h}_{\hat{\boldsymbol{x}}} \in \mathbb{R}^d$, where $d$ is the dimension of hidden states. Then the MLM head of $\mathcal{M}$ can operate on $\boldsymbol{h}_{\hat{\boldsymbol{x}}}$ to calculate each word $v$'s probability in the vocabulary being filled in [MASK] $P_{\mathcal{M}}(\,[\text{MASK}] = v|\hat{\boldsymbol{x}})$. We let $\mathcal{V}_y$ to represent the subset of $\mathcal{V}$ which is connected with a unique label $y$, $\cup_{y \in \mathcal{Y}} \mathcal{V}_y = \mathcal{V}$. Finally, the probability distribution over the label $y$ is calculated as:

$$P(y|\boldsymbol{x}) = g\left(P_{\mathcal{M}}(\,[\text{MASK}] = v|\mathcal{T}(\boldsymbol{x}))|v \in \mathcal{V}_y\right), \tag{2}$$

where $g$ refers to the function converting the probability of label words to the probability of classes.

## 3    RETROPROMPT: Retrieval-augmented Prompt Learning

We introduce a simple and general retrieval-augmented framework for prompt learning, named RETROPROMPT, whose basis is the dense retriever (§3.1) with an open-book knowledge-store to decouple knowledge from memorization. As shown in Figure 2, RETROPROMPT consists of three components: retrieval of neural demonstration for enhancing input (§3.2), the $k$NN guided training (§3.3) and the $k$NN-based probability for *cloze-style* prediction (§3.4).

### 3.1   Dense Retriever

**Open-book Knowledge-store**    The first step of our proposed framework is to build a knowledge-store for retrieval that can decouple from memorization and captures the semantics of the instance from the training set $\mathcal{C}$. Specifically, we leverage the encoder to embed instance representation over the $\mathcal{C}$ to construct the knowledge-store. Given the $i$-th example $(\boldsymbol{c}_i, y_i)$ in the training data $\mathcal{C}$, we obtain the key-value pair $(\boldsymbol{h}_{\hat{\boldsymbol{c}}_i}, v_i)$, in which $\hat{\boldsymbol{c}}_i = \mathcal{T}(\boldsymbol{c}_i)$, $\boldsymbol{h}_{\hat{\boldsymbol{c}}_i} \in \mathbb{R}^d$ is the embedding of the [MASK] token in the last layer of the PLM, and $v_i = f(y_i)$ denotes the label word of the $i$-th example. Compared with kNN-LM [26] that constructing with sliding generative corpus and tokens, our knowledge-store is more suitable for prompt learning. We store all pairs $(\boldsymbol{h}_{\hat{\boldsymbol{c}}}, v)$ in a key-value datastore $(\mathcal{K}, \mathcal{V})$ where $\boldsymbol{h}_{\hat{\boldsymbol{c}}}$ serves as *key* and $v$ as *value* as follows:

$$(\mathcal{K}, \mathcal{V}) = \{(\boldsymbol{h}_{\hat{\boldsymbol{c}}_i}, v_i) \mid (\boldsymbol{c}_i, y_i) \in \mathcal{C}\} \tag{3}$$

The knowledge-store maye be flexible to edit, add or delete any instances and can be asynchronously updated during the training procedure. Note that our knowledge-store is constructed from few-shot trainsets in the corresponding few-shot settings rather than the whole available training data.

**Efficient Searching**    Considering that the size of the training data $\mathcal{C}$ can be enormous, we must ensure an efficient retrieval process. As shown in the above creation of open-book knowledge-store,

we can build the matrix $\mathbf{D} \in \mathbb{R}^{|\mathcal{C}| \times d}$ as the index of training examples. Given a query set $Q$, we first encode each query example with template mapping function $\mathcal{T}(\cdot)$ to get a set of prompt-based query vectors $\boldsymbol{h}_{\hat{q}}$ for retrieval augmentation on the fly. Then, we utilize query vectors to search for the closest examples over the index $\mathbf{D}$ via maximum inner product search (MIPS). For the retrieval process, we choose FAISS [21] to query the open-book knowledge-store efficiently. FAISS is an excellent open-sourced toolkit for fast nearest neighbor retrieval.

**Asynchronous Refresh of the Knowledge-store**    Since the neural demonstration may lead to the variable contextual representation of instance as the parameters of the PLM are continually updated, we thus propose to "refresh" the index of retrieval by asynchronously re-embedding and re-indexing all embeddings in an open-book knowledge-store every $j$ training epochs [4]. In § 4.6, we empirically demonstrate that this procedure results in performance improvement.

## 3.2    Retrieval of Neural Demonstration

To enhance the PLMs with the ability to learn by analogy through the knowledge-store, we further propose neural demonstrations that can be concatenated with input instance at the embedding layer to improve the generalization ability of our RETROPROMPT. For the $t$-th query instance $\boldsymbol{q}_t$, we first utilize prompt-based representation $\boldsymbol{h}_{\hat{q}_t}$ to query the cached representations of open-book knowledge-store. Then we retrieve $m$ nearest neighbors $\{\{\boldsymbol{c}_1^{(1)}, ..., \boldsymbol{c}_m^{(1)}\}, ..., \{\boldsymbol{c}_1^{(L)}, ..., \boldsymbol{c}_m^{(L)}\}\}$ of $\boldsymbol{q}_t$ for each class, where the superscript $L$ denotes the total number of the classes and the $\boldsymbol{c}_i^{(l)}$ is retrieved as the $i$-th nearest neighbor in the $l$-th class. After the model retrieves the Top-$m$ candidates for each class, their corresponding representation $\boldsymbol{h}_{\hat{c}_i}^{(l)}$ and label word $v^{(l)}$ from knowledge-store will be incorporated into the encoder to act as a demonstration learning. Since the $\boldsymbol{h}_{\hat{c}_i}^{(l)}$ is already vector, we intuitively aggregate the $m$ neighbor vectors for each class according to their similarity and incorporate the demonstration into the input representation of $\hat{\boldsymbol{x}}$ after the word embedding layer of the $\mathcal{M}$ as follows:

$$\mathcal{I} = e(\hat{\boldsymbol{x}}) \oplus [\sum_{i \in [1:m]} \alpha_i^{(1)} \boldsymbol{h}_{\hat{c}_i}^{(1)}, e(v^{(1)})] \oplus ... \oplus [\sum_{i \in [1:m]} \alpha_i^{(L)} \boldsymbol{h}_{\hat{c}_i}^{(L)}, e(v^{(L)})]; \alpha_i^{(l)} = \frac{e^{\boldsymbol{h}_{\hat{q}} \cdot \boldsymbol{h}_{\hat{c}_i}^{(l)}}}{\sum_{i \in [1:m]} e^{\boldsymbol{h}_{\hat{q}} \cdot \boldsymbol{h}_{\hat{c}_i}^{(l)}}} \quad (4)$$

where $e(\cdot)$ represents the word embedding layer of $\mathcal{M}$, $\oplus$ denotes the concatenation of input sequences, $\alpha_i^{(l)}$ is the softmax score for the $i$-th retrieval belonging to $l$-th class label to denote their relevance with $\hat{q}$, and $\mathcal{I}$ is the sequence features for inputting the next layer of PLM. As shown in the above equation, we encode demonstration representation with the weighted sum of the retrieval representation. Thus, retrieval scores are directly used in the final representation, making the framework differentiable. To this end, we denote this style of demonstration as *neural demonstration*, significantly different from prior work of *discrete demonstration* [12].

**Neural vs. Discrete Demonstration** Compared with prior discrete demonstrations described in [12, 35, 49, 28], retrieving weighted neural demonstrations from the knowledge-store to augment prompt learning has advantages in the following three major aspects: (1) neural demonstrations could be more tolerant of the model's maximum input length than discrete demonstrations, while the discrete demonstration is usually not suitable for multi-class classification tasks due to the limitation of input length, such as relation extraction, etc. (2) the model needs to deal with large retrieval tokens for discrete demonstration, making it time-consuming and computationally intensive to perform cross-attention operations due to the quadratic attention complexity. In contrast, dealing with much shorter instance representations as neural demonstrations unleashes the potential of cross-attention and accelerates the inference. (3) when sampling examples based on the similarity between instances, our *cloze-style* contextual representation is more informative and consistent than the contextual representation from `[CLS]` of Sentence-BERT [47] (adopted in LM-BFF).

## 3.3    Retrieve $k$NN for Guiding Training

Eager learners (e.g., PLMs) are optimized to learn a global function that maps from the text to semantic label space. Lazy learners such as $k$-nearest neighbor classifiers, on the contrary, aims to

---

[4]Specifically, we refresh the knowledge-store for each epoch in our experiments.

approximating the neighborhoods around those test examples [2]. Since $k$NN can easily predict for each encountered query instance based on pre-trained representation without an extra classifier, it is intuitively to leverage the $k$NN's classification results as the **prior external knowledge** to guide the PLMs' parameters attending to hard examples (hard samples usually refer to atypical samples) during the training process (also referred as $k$NN-train for the abbreviation). Particularly, our intuition is to differentiate between easy and hard examples according to the prediction of $k$NN. Given the $t$-th query instance $\boldsymbol{q}_t$, we leverage the $\boldsymbol{h}_{q_t}$ querying the open-book knowledge-store $(\mathcal{K}, \mathcal{V})$ to retrieve the $k$-nearest neighbors $\mathcal{N}$ of $\boldsymbol{q}_t$ according to a similarity function $d(\cdot, \cdot)$, where $d(\cdot, \cdot)$ typically adopt the inner product similarity. Then, we compute the distribution over neighbors according to the softmax of their similarities and aggregate probability mass for each label word across its occurrences in the retrieved targets:

$$P_{k\text{NN}}(y \mid \boldsymbol{q}_t) \propto \sum_{(\boldsymbol{c}_i, y_i) \in \mathcal{N}} \mathbb{1}_{y=y_i} \exp\left(d\left(\boldsymbol{h}_{\hat{q}_t}, \boldsymbol{h}_{\hat{c}_i}\right)\right). \tag{5}$$

Given the probability $p_{k\text{NN}}$ of the query instance $\boldsymbol{q}_t$ being predicted as the **gold class** (also as the probability value of the gold class in the $P_{k\text{NN}}$), we propose to retrieve the $k$NN for guiding the training process of prompt learning. The $k$NN guider reweights the cross-entropy loss $\mathcal{L}_{CE}$ by adjusting the relative loss for the correctly-classified or misclassified instances identified by $k$NN, respectively. Specifically, we apply the negative log-likelihood as the modulating factor $F(p_{k\text{NN}})$. The final loss $\mathcal{L}$ is defined as:

$$F(p_{k\text{NN}}) = -\log\left(p_{k\text{NN}}\right), \quad \mathcal{L} = \left(1 + \beta F(p_{k\text{NN}})\right) \mathcal{L}_{CE}, \tag{6}$$

where $\beta$ denotes a scalar to determine the proportion of each loss term. Note that $p_{k\text{NN}}$ is computed using the *leave-one-out* distribution on the training set due to the fact that each example in the training set cannot retrieve itself. The motivation of modulating factor is inspired by Focal-loss [34], while we focus on exploit leveraging k-NN's results for calibrating the training of LMs.

### 3.4 $k$NN based probability for *Cloze-style* Prediction

Apart from the neural demonstration on the input side and $k$NN guided training process (also referred as $k$NN-test for the abbreviation), we further present $k$NN based probability for *Cloze-style* prediction on the inference process, providing the PLM ability to retrieve nearest neighbors for decisions rather than making predictions only based on memorized parameters. Given the non-parametric $k$ nearest neighbor distribution $P_{k\text{NN}}$ of the query instance $\boldsymbol{q}_t$ being predicted as $y$, we follow [13, 26, 16] to reformulate the $P(y \mid \boldsymbol{q}_t)$ by interpolating the $P_{k\text{NN}}$ with the already-trained base PLM's MLM prediction $P_\mathcal{M}$ using parameter $\lambda$ to produce the final probability of the label:

$$P(y \mid \boldsymbol{q}_t) = \lambda P_{k\text{NN}}(y \mid \boldsymbol{q}_t) + (1 - \lambda) g\left(P_\mathcal{M}(\texttt{[MASK]} = v | \mathcal{T}(\boldsymbol{q}_t))\right). \tag{7}$$

Different from $k$NN-LM [26, 16] that mainly retrieve tokens to augment the language modeling, we focus on leveraging prompt-based kNN's distribution for reference at test time, which can unlock the model prediction process as an *open-book* examination for prompt learning.

## 4 Experiments

### 4.1 Datasets and Baselines

**Datasets** We evaluate RETROPROMPT on several types of natural language understanding tasks, including single sentence classification tasks (SST-2 [54], MR [43], and CR [19]) and sentence pair classification tasks (MNLI [58], QNLI [46], and QQP[5]). To further evaluate the effectiveness of the proposed approach with multi-class classification, we also conduct experiments on the information extraction tasks, including FewNERD [8], SemEval 2010 Task 8 (SemEval) [17], and TACRED [61]. The detailed statistics of the datasets are shown in Appendix A.

**Baselines** We compare with LM-BFF [12] for single sentence and sentence pair classification tasks and adopt SOTA prompt learning model KnowPrompt [6] as the baseline for information extraction tasks. Note that the discrete demonstration method cannot be applied to multi-class classification tasks due to the input length limitations; thus, we leave out the experimental table about the results of KnPr (D-demo). We also compare our RETROPROMPT with the knowledge-enhanced prompt learning

---

[5]`https://www.quora.com/q/quoradata/`.

Table 1: Results across 9 NLU datasets in the few-shot and zero-shot setting. We report mean (and standard deviation) results over five different few-shot splits. "D-demo" refers to discrete demonstration, and "KnPr" is the abbreviation of KnowPrompt. LOTClass [41] is the SOTA model in unsupervised text classification with self-training. † donates the model uses **extra knowledge** and ♣ means they **train** the PLM on the whole unlabeled trainset, while we and the other baselines only leverage the vanilla PLM to test without training. The average scores with * denote that we reuse the results of the "non-demo" version of the related model to fill in the default values.

| St. | Model | Single Sentence | | | Sentence Pair | | | Model | Information Extraction | | | Avg. |
|---|---|---|---|---|---|---|---|---|---|---|---|---|
| | | SST-2 (acc) | MR (acc) | CR (acc) | MNLI (acc) | QNLI (acc) | QQP (F1) | | FewN (acc) | SemEval (acc) | TACRED (F1) | |
| 16 | FT | 81.4 (3.8) | 76.9 (5.9) | 75.8 (3.2) | 45.8 (6.4) | 60.2 (6.5) | 60.7 (4.3) | FT | 52.7 (2.2) | 66.1 (1.2) | 25.8 (2.8) | 60.6 |
| | LM-BFF (man) | 91.6 (1.2) | 87.0 (2.0) | 90.3 (1.6) | 64.3 (2.5) | 64.6 (5.4) | 65.4 (5.3) | KnPr | 65.3 (1.1) | 80.9 (2.5) | 33.2 (2.0) | 71.4 |
| | LM-BFF (D-demo) | 91.8 (1.2) | 86.6 (1.8) | 90.2 (1.4) | 64.8 (2.3) | 69.2 (5.4) | 68.2 (3.2) | KnPr (D-demo) | — | — | — | 72.2* |
| | KPT † | 90.3 (1.6) | 86.8 (1.8) | 88.8 (3.7) | 61.4 (2.1) | 61.5 (2.8) | 71.6 (2.7) | KPT † | 65.9 (1.5) | 78.8 (2.1) | 32.8 (1.7) | 70.9 |
| | **Ours** | **93.9 (0.4)** | **88.0 (0.8)** | **91.9 (0.7)** | **71.1 (1.8)** | **71.6 (1.8)** | **74.0 (2.0)** | **Ours** | **67.3 (0.9)** | **81.5 (1.3)** | **40.7 (0.7)** | **75.6** |
| 4 | FT | 60.2 (2.8) | 57.6 (1.4) | 66.4 (5.5) | 35.0 (0.3) | 54.2 (3.9) | 52.8 (4.7) | FT | 32.7 (2.9) | 38.8 (2.0) | 14.7 (2.8) | 45.8 |
| | LM-BFF (man) | 90.7 (0.8) | 85.2 (2.8) | 89.9 (1.8) | 51.0 (2.5) | 61.1 (6.1) | 48.0 (4.9) | KnPr | 52.5 (1.5) | 58.4 (3.7) | 28.8 (2.5) | 62.8 |
| | LM-BFF (D-demo) | 90.2 (1.5) | 85.5 (2.1) | 89.7 (0.6) | 56.1 (1.0) | 61.7 (7.6) | 63.2 (5.6) | KnPr (D-demo) | — | — | — | 65.1* |
| | KPT † | 88.2 (5.7) | 83.4 (1.5) | 87.2 (2.5) | 53.7 (2.7) | 59.2 (2.8) | 54.9 (7.9) | KPT † | 58.8 (2.2) | 57.2 (3.2) | 27.5 (2.2) | 63.3 |
| | **Ours** | **91.5 (1.8)** | **87.4 (0.5)** | **91.4 (0.6)** | **57.6 (5.5)** | **62.2 (6.0)** | **66.1 (4.1)** | **Ours** | **60.9 (1.9)** | **59.2 (3.0)** | **32.1 (2.0)** | **67.6** |
| 0 | LOTClass♣ | 71.8 | 81.7 | 50.1 | 50.4 | 36.5 | 55.9 | LOTClass♣ | 11.5 | 9.8 | 2.5 | 41.1 |
| | FT | 49.1 | 50.0 | 49.8 | 34.4 | 49.5 | 31.6 | FT | 10.0 | 6.2 | 0.5 | 31.2 |
| | LM-BFF (man) | 83.5 | 80.3 | 78.4 | 49.7 | 50.5 | 49.7 | KnPr | 15.9 | 10.3 | 2.3 | 46.7 |
| | LM-BFF (D-demo) | 82.9 | 80.7 | **81.4** | 52.2 | 53.5 | 44.0 | KnPr (D-demo) | — | — | — | 47.0* |
| | KPT † | 78.4 | 81.9 | 71.4 | 37.1 | 55.3 | 47.5 | KPT † | 24.6 | 11.6 | 0.8 | 45.7 |
| | **Ours** | **86.8** | **83.5** | 79.7 | **53.7** | **56.2** | **56.7** | **Ours** | **41.3** | **12.2** | **2.8** | **52.5** |

method KPT [20] since KPT leverages the external knowledge base for enhancing prompt learning while we focus on utilizing internal trainsets as a knowledge-store. You can refer to Appendix B for the detailed introduction of baseline methods.

## 4.2 Evaluation protocols and details

The experiments are implemented on 1 NVIDIA V100 and utilize Pytorch [44] as the base library. We adopt RoBERTa$_{large}$ [38] as the PLM and employ AdamW as the optimizer for all experiments. To mitigate the influence of diverse templates, we conduct baselines and RETROPROMPT with the same templates for each dataset. We list the specific experimental settings and tuning retrieve parameters in Appendix C and D. As for few-shot and zero-shot experiments, we leverage different settings, respectively.

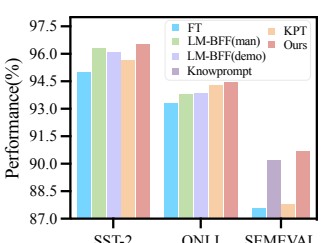

Figure 3: Performance on fully-supervised datasets.

**Few-shot Setting.** We follow the few-shot setting of LM-BFF [12] to conduct 4-shot and 16-shot experiments and evaluate the average performance with a fixed set of seeds, $\mathcal{S}_{seed}$, across several different sampled $\mathcal{D}_{train}$ for each task. Note that our knowledge-store is constructed with the **few-shot training set** in this setting.

**Zero-shot Setting**[6]**.** We leverage vanilla RoBERTa$_{large}$ for all baselines (except LOTClass [41]) to directly inference on the test set. To take advantage of retrieval mechanism, RETROPROMPT follows LOTClass [41] to utilize **unlabeled** trainsets for retrieval. Specifically, we take the vanilla RoBERTa$_{large}$ to tag the pseudo labels on unlabeled trainset and create the open-book knowledge-store with the unlabeled trainsets and pseudo labels. Lastly, RETROPROMPT make predictions on the test set based on the constructed datastore **without tuning any of the model parameters**.

## 4.3 Experimental Results

**Few-shot Results.** As shown in Table 1, we find RETROPROMPT consistently outperforms baseline method LM-BFF and KnowPrompt, both in 4-shot and 16-shot experiments. Especially for information extraction tasks with multiple classes, discrete demonstrations cannot be applied to the input due to the limited input sequence length, while our neural demonstration can also work and achieves

---

[6]Note that it is not a strict zero-shot sense.

improvement on these multi-class datasets. Moreover, RETROPROMPT obtain better performance compared with KPT. Compared with KPT with external knowledge, we only focus on referencing the internal few-shot trainsets without visiting the external knowledge base. Besides, we observe that RETROPROMPT has a relatively lower standard deviation than the baselines. The reason may lie that the retrieval mechanism can compensate for instabilities in parametric predictions.

**Zero-shot Results.** From Table 1, we also observe that RETROPROMPT achieves improvements in the zero-shot setting. Another notable point is that RETROPROMPT performs even better than KPT in the zero-shot setting, revealing that exploring own data to decouple knowledge from memorization has more potential than leveraging external knowledge. Moreover, we achieve superior performance to LOTClass even though we utilize the vanilla RoBERTa$_{\text{large}}$ without any training.

**Fully-supervised Results.** As shown in Figure 3, the experiments in fully-supervised settings with long-tail distribution illustrate that RETROPROMPT achieves improvement compared with baselines. This indicates that our retrieval mechanism extends the LM's ability to learn hard examples in the fully-supervised datasets.

Table 2: Results of model generalization to new domains.

| Model | Source | Target Domain | |
|---|---|---|---|
| | 16-shot MR | SST-2 | CR |
| FT | 76.9 | 71.4 | 64.7 |
| LM-BFF (man) | 87.0 | 88.9 | 86.9 |
| LM-BFF (D-demo) | 86.6 | 89.3 | 87.5 |
| KPT | 86.8 | 86.8 | 86.7 |
| **RETROPROMPT** | **88.0** | **91.4** | **88.8** |
| | 16-shot QQP | MRPC | RTE |
| FT | 60.7 | 43.7 | 48.0 |
| LM-BFF (man) | 65.4 | 20.9 | 65.5 |
| LM-BFF (D-demo) | 68.2 | 38.8 | 66.2 |
| KPT | 71.6 | 42.3 | 65.8 |
| **RETROPROMPT** | **74.0** | **49.4** | **67.3** |

## 4.4 Model Generalization to New Domains

The scarce data may bring the overfitting problem for the lots of memory parameters of PLMs, even though prompt learning. Thus, we conduct cross-domain experiments to validate the generalization of our RETROPROMPT. Specifically, we utilize the model trained on the source datasets and directly test on the other target datasets. From Table 2, we can find that our method consistently outperforms baselines. This finding illustrates that RETROPROMPT achieves great model generalization to new domains.

## 4.5 Analysis of Memorization

It is necessary and interesting to further explore the memorization mechanism to help us better understand the utility of retrieval for memorization in NLP.

**Definition of Memorization Measurement.** Inspired by the idea of [10] in the computer vision area, we define *memorization measures* as to how the classification varies when a training instance $z$ is deleted from the trainset. We follow [27, 62] to define and derive the memorization score for a training instance $z$ as follows:

$$\mathcal{S}_{\text{delate}}(z) \overset{\text{def}}{=} -\frac{dP(y|\boldsymbol{x};\hat{\theta}_{\xi,-z})}{d\xi}\bigg|_{\xi=0} = -\nabla_\theta P(y|\boldsymbol{x};\hat{\theta})^\top \frac{d\hat{\theta}_{\xi,-z}}{d\xi}\bigg|_{\xi=0} = -\nabla_\theta P(y|\boldsymbol{x};\hat{\theta})^\top H_{\hat{\theta}}^{-1}\nabla_\theta \mathcal{L}(\boldsymbol{z},\hat{\theta}),$$

(8)

where $\hat{\theta}_{\xi,-z}$ denotes the parameters trained with the instance $z$ down-weighted by $\xi$, $\hat{\theta}$ refers to the parameters of the model trained with all instances and $H_{\hat{\theta}} = \frac{1}{n}\sum_{i=1}^{n}\nabla_\theta^2 \mathcal{L}(z_i,\hat{\theta})$. Thus $\mathcal{S}_{\text{delate}}(z)$ refers to the amount of change of $P(y|x;\theta)$ when the instance $z$ is down-weighted by $\xi$.

**Top-memorized Instances: Typical or Atypical?** Since the SST-2 dataset provides the annotations of phrase-level sentiment polarity labels, we adopt SST-2 to analyze the memorization by judging the atypical of an instance by checking the percentage of positive phrases. We achieve such statistics from SST-2 and observe that a typical positive instance has a relatively high percentage of positive phrases, and a typical negative instance should have a relatively low percentage of positive phrases. Based on the above observation, we apply the memorization score defined in Eq. 8 to select Top-10% and Bottom-10% memorized instances from the trainset and collect the average percentage of positive phrases in these instances.

As shown in Table 3, we can conclude following findings: (1) **The PLM tends to give atypical samples deeper memory attention.** Specifically, no matter LM-BFF or our method, the top-

Table 3: The upper part shows the average percentage of *positive phrases* over different memory groups of positive/negative instances. The lower part denotes the mean values of memorization score on the SST-2 dataset.

| Mem Group | Negative | | | Postive | | |
|---|---|---|---|---|---|---|
| | FT | LM-BFF | OURS | FT | LM-BFF | OURS |
| Top-10% | 34.29 | 32.78 | 30.23 | 68.75 | 69.71 | 75.67 |
| ALL | | 23.40 | | | 86.39 | |
| Bottom-10% | 17.63 | 16.25 | 14.42 | 95.92 | 95.08 | 94.53 |

| | FT | LM-BFF | OURS |
|---|---|---|---|
| MEM SCORE | 4.597 | 0.121 | 0.032 |

10% memorized negative instances have a higher percentage of positive phrases than the average percentage of positive phrases of all negative instances. 2) LM-BFF has lower memorization scores on hard samples than fine-tuning. We think it owns to **prompt learning can help PLMs recall what they learned from pre-training without strengthening memory for downstream data.** 3) RETROPROMPT further has lower average memorization scores than fine-tuning and LM-BFF, which illustrates that our method is less memory dependent. This result may be attributed to **decoupling knowledge from memorization through retrieval to alleviating the rote of PLMs.**

**Case Analysis.** As shown in Table 6, we manually list the bottom-ranked and top-ranked training instances of SST-2 according to our model. It reveals that the top-ranked memorized instances seem to show universal opinions indirectly. Thus, we inspect them as atypical/hard for sentiment classification. While those instances with 0 memorization scores are straightforward to show their opinion for sentiment classification, representing the typical instance. Note that $F(p_{kNN})$ is defined to represent the difficulty of the sample discriminated by $k$NN distribution. And the Table 6 also shows that $F(p_{kNN})$ indeed reflect atypicality of examples, which validates the effectiveness of the $k$NN guided training.

Table 4: Detailed ablation experiments in few-shot settings. "N-demo" donates the neural demonstration, and "refresh" refers to the asynchronous refresh of the knowledge-tore.

| Model | 16-shot | | | | |
|---|---|---|---|---|---|
| | SST-2 | CR | MNLI | QQP | TACRED |
| **OURS** | **93.9** | **91.9** | **71.1** | **74.0** | **40.7** |
| w/o $k$NN-test | 93.2 | 91.2 | 70.4 | 73.0 | 38.2 |
| w/o $k$NN-train | 92.0 | 90.2 | 68.8 | 71.3 | 36.5 |
| w/o N-demo | 92.4 | 91.0 | 70.1 | 72.7 | 37.9 |
| w/o refresh | 93.5 | 91.5 | 70.7 | 73.6 | 39.9 |

## 4.6 Ablation Study

**Component Ablation.** As shown in Table 4, the performance of component ablation experiments with four variants has a clear drop, which validates the power of our retrieval component. We also find that neural demonstration and $k$NN-train have more improvement in the few-shot setting than $k$NN-test. Note that $k$NN-test is similar to $k$NN-LM [26, 16] and the results reveals that simply incorporate $k$NN in the test process of prompt learning has little influence in a few-shot setting.

**Key Representation and $k$NN Acquisition.** We study the effect of using different representations of the key in the knowledge-store. We experiment with two types of representations: (1) prompt-based representation, which is the default setting, and (2) [CLS] based representation of current LM. We also experiment with two types of calculation of $k$NN distribution: (1) representation based similarity score (refer as rep-similar), which is the default setting, and (2) BM25 based score , which calculates the correlation score between the query and each key examples with BM25 [48] algorithm. Results in Table 5 show that using prompt-based representations for key and representation based similarity scores for $k$NN leads to the best performance. It suggests that prompt learn better representations for context similarity and the representation similarity based $k$NN distribution is better than BM25 based scores.

Table 5: Performance on 16-shot CR and TACRED with different representations of key and calculate function of $k$NN distribution.

| Key Repres. | $k$NN Acq. | CR | TAC. |
|---|---|---|---|
| Prompt | Rep-similar | 91.9 | 40.7 |
| [CLS] | Rep-similar | 89.0 | 37.2 |
| Prompt | BM25 | 89.5 | 38.8 |
| [CLS] | BM25 | 88.7 | 36.1 |

Table 6: Case examples of Top-3 and Bottom-3 memorized instance of ours from trainset of SST-2.

| Negative | | | Positive | | |
|---|---|---|---|---|---|
| **Content** | **Mem** | $F(p_{k\text{NN}})$ | **Content** | **Mem** | $F(p_{k\text{NN}})$ |
| Although god is great addressed interesting matters of identity and heritage, it's hard to shake the feeling that it was intend to be a different kind of film. | 0.066 | 1.17 | A b-movie you can sit through, enjoy on a certain level and then forget. | 0.020 | 0.18 |
| A standard police-oriented drama that, were it not for deniro's participation, would have likely wound up a tnt original. | 0.011 | 1.48 | A film that will be best appreciated by those willing to endure its extremely languorous rhythms, waiting for happiness is ultimately thoughtful without having much dramatic impact. | 0.010 | 0.43 |
| A hit and miss affair, consistently amusing but not as outrageous or funny as cho may have intended or as imaginative as one might have hoped. | 0.010 | 2.74 | What's invigorating about is that it doesn't give a damn. | 0.003 | 0.06 |
| It's a loathsome movie, it really is and it makes absolutely no sense. | 0.00 | 0.00 | A fun family movie that's suitable for all ages– a movie that will make you laugh, cry and realize, 'it's never too late to believe in your dreams.' | 0.00 | 0.00 |
| It is that rare combination of bad writing, bad direction and bad acting – the trifecta of badness. | 0.00 | 0.00 | It's a cool event for the whole family. | 0.00 | 0.00 |
| This thing is virtually unwatchable. | 0.00 | 0.00 | Good fun, good action, good acting, good dialogue, good pace, good cinematography. | 0.00 | 0.00 |

## 5   Related Work

**Retrieval-enhanced PLMs.**   Our pipeline is partly inspired by discrete demonstration methods such as [12, 35, 28, 29] that retrieves few training examples in a natural language prompt, while we propose neural demonstration for enhancing the input to alleviate the limitations of input length. Another line researches of retrieval augmentation [14, 23, 32, 49, 3] retrieve useful information from a external knowledge corpus (e.g., Wikipedia) for a particular task (e.g., an open-domain question). Unlike these works, we focus on retrieving examples from the internal training data. Besides, semi-parametric methods [26, 16, 25, 24, 1, 42] have risen to leverage $k$-nearest neighbor classifier, a classic non-parametric algorithm that makes the prediction based on representation similarities, to enhance pre-trained language models in various tasks However, unlike these models using nearest neighbors only for augmenting the process of prediction, we aim to develop a comprehensive retrieval mechanism for input, training and test process.

**Prompt learning for PLMs.**   With the birth of GPT-3 [4], prompt learning [36] has recently arisen to fill the gap between masked LM objective of PLMs and downstream fine-tuning objective. Prompt learning has achieves very impressive performance on various tasks, such as text classification [51, 53], named entity recognition [5, 40], relation extraction [15, 6], event extraction [18, 60], machine translation [55] and language generation [52], especially under the setting of few-shot learning. Moreover, continuous prompts have also been proposed [33, 30, 37] to reduce prompt engineering, which directly appends a series of learnable continuous embeddings as prompts into the input sequence. Our work is orthogonal to previous prompt learning approaches, which aim to optimize prompts, while we focus on the systematic study of retrieving related examples from training data to enhance prompt learning.

## 6   Conclusion and Future Work

We propose RETROPROMPT that decouples knowledge from memorization by introducing retrieval augmentation to further improve the generalization ability of prompt learning on the input side and the whole process of model training and prediction. RETROPROMPT, is a straightforward yet effective retrieval method that combines both neural demonstrations, $k$NN guider for training and prediction. Our extensive results show that it outperforms other demonstration-enhanced prompt methods and knowledge-enhanced prompt methods in few-shot, zero-shot and fully-supervised settings. Analyzing the essence of memorization validates the effectiveness of decoupling knowledge from memorization. Interesting future directions include: 1) apply to other tasks, such as QA and NLG, 2) explore the noise data mining for unsupervised learning, 3) further improve the retrieve efficiency for large datasets, etc.

## Acknowledgments

We want to express gratitude to the anonymous reviewers for their kind comments. This work was supported by National Natural Science Foundation of China (No.62206246, 91846204 and U19B2027), Zhejiang Provincial Natural Science Foundation of China (No. LGG22F030011), Ningbo Natural Science Foundation (2021J190), and Yongjiang Talent Introduction Programme (2021A-156-G).

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
