# Appendix of *"Decoupling Knowledge from Memorization:* Retrieval-augmented Prompt Learning"

**Xiang Chen**[1,2]*, **Lei Li**[1,2]*, **Ningyu Zhang**[1,2]†, **Xiaozhuan Liang**[1,2], **Shumin Deng**[1,2],
**Chuanqi Tan**[3], **Fei Huang**[3], **Luo Si**[3], **Huajun Chen**[1,2]†
[1]Zhejiang University & AZFT Joint Lab for Knowledge Engine, China
[2]Hangzhou Innovation Center, Zhejiang University, China
[3]Alibaba Group, China
{xiang_chen, leili21, zhangningyu, liangxiaozhuan, 231sm, huajunsir}@zju.edu.cn,
{chuanqi.tcq, f.huang, luo.si}@alibaba-inc.com

## A   Datasets and Templates

In this section, we introduce the datasets as shown in Table 1 and list the templates we use in experiments as follows.

Table 1: Detailed dataset statistics.

| Dataset | Type | # Class | Test Size |
|---------|------|---------|-----------|
| SST-2 | Sentiment | 2 | 872 |
| MR | Sentiment | 2 | 2,000 |
| CR | Sentiment | 2 | 2,000 |
| MNLI | NLI | 3 | 9,815 |
| QNLI | NLI | 2 | 5,463 |
| QQP | Paraphrase | 2 | 40,431 |
| FewNERD | Entity Typing | 66 | 96,901 |
| SemEval | Relation Extraction | 19 | 2,717 |
| TACRED | Relation Extraction | 42 | 15,509 |

**SST-2, MR, CR**.   For the single sentence classification tasks, we follow the LM-BFF [4] to design the templates:

$$T(\mathbf{x}) = \texttt{[CLS]}\, \mathbf{x}\ \text{It was [MASK].}$$

We set Verbalizer: (great/terrible) → (positive/negative) for SST-2 MR and CR. For the Yahoo dataset, we assign the Verbalizer following the original labels.

**MNLI, QNLI, QQP**.   For the sentence pair classification tasks, we follow LM-BFF [4] to set Verbalizer: (Yes/Maybe/No) → (entailment/neutral/contradiction), and define the following templates:

$$T(\mathbf{x_1}, \mathbf{x_2}) = \texttt{[CLS]}\,\mathbf{x_1}?\,\texttt{[MASK]}, \mathbf{x_2}$$

**FewNERD, SemEval, TACRED**.   FewNERD, SemEval and TACRED are datasets for information extraction, which require inserting the entity into the template. Therefore, we follow [3] and [2] to define the template and verbalizers.

---

\* Equal contribution.
† Corresponding Author.

36th Conference on Neural Information Processing Systems (NeurIPS 2022).

## B Compared Baselines

In this subsection, we introduce the baselines we compare with and re-produce them under the same settings with their open-source codes.

LM-BFF uses several other tricks, such as prompt ensemble, while KPT utilizes tremendous external knowledge. We do not use any of these tricks and external knowledge since we get the most out of the data to decouple part of knowledge from parametric memorization. Our RETROPROMPT mechanism is orthogonal to other methodological improvements of prompt-tuning (such as continuous prompt in P-tuning [6] and DART [8] ) and can be combined with other prompt-tuning methods in future work.

**Fine-tuning (FT).** The traditional fine-tuning method regard the hidden embedding of `[CLS]` token of the PLM as the representation of the sentence and then feeds them into a classification layer to make predictions.

**LM-BFF.** LM-BFF [4] is a typical prompt-tuning method wrapping an input sentence into a handcrafted template. Here we re-produce LM-BFF based on their open-source codes [3] with the same manual prompts as RETROPROMPT for a fair comparison.

**LM-BFF (+Demo).** This approach is the above LM-BFF [4] combined with the demonstration [1]. Different from RETROPROMPT, it uses examples of natural language as demonstrations, which is restricted by the input length of the language model. Thus, LM-BFF (+demo) is not suitable for multi-class classification tasks.

**KnowPrompt.** KnowPrompt [2] is a SOTA prompt-tuning method for relation extraction tasks with multiple classes. We apply our RETROPROMPT over KnowPrompt on information extraction tasks for comparison, aiming to verify the broad applicability of our method.

**Incorporating Knowledge into Prompt (KPT).** KPT [5] focuses on incorporating external knowledge into the verbalizer by refining the expanded label word space to improve and stabilize prompt-tuning, which is a solid baseline for comparison. We follow their public codes[4] to conduct experiments in the same setting for a fair comparison.

**LOTClass.** LOTClass [7] is the SOTA method in unsupervised text classification that utilizes the PLM to extract the label-related words from the whole unlabeled training corpus. Then it leverages the Masked Category Prediction task to **train** on the unlabeled corpus with pseudo labels.

## C Experimental Settings

We report the hyper-parameters in Table 2. Most of the hyper-parameters are the default parameters of LM-BFF[5].

Table 2: Hyper-parameter settings.

| Hyper-parameter | Value |
|---|---|
| maximum sequence length | {128, 256} |
| max training step | 800 |
| evaluation step | 80 |
| learning rate | {1e-5, 2e-5, 5e-5} |
| batch size | {2, 4, 8} |
| adam epsilon | 1e-8 |

## D Tuning Retrieve Parameters

The final distribution of the label is affected by the hyperparameters of $\beta$, $k$ and $\lambda$ when conducting $k$NN-train and $k$NN-test. Thus, we provide insight into the effect of $\beta$, $k$ and $\lambda$ on the final results.

$\beta$ **varies:** Figure 1(a) shows the performance of the model when the $\beta$ increases and reveals that the model performs worse as the $\beta$ increases on a 16-shot CR dataset. This finding indicates that a

---

[3] https://github.com/princeton-nlp/LM-BFF
[4] https://github.com/ShengdingHu/KnowledgeablePromptTuning
[5] https://github.com/princeton-nlp/LM-BFF

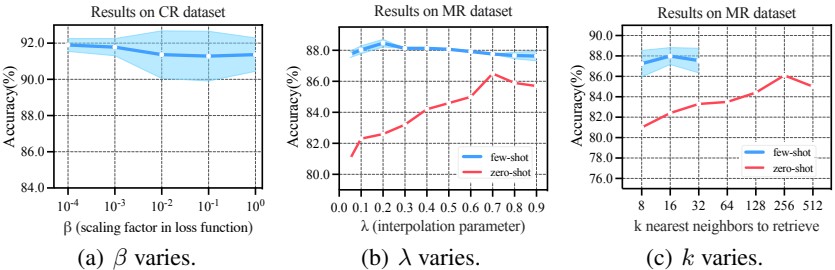

Figure 1: Effect of the hyperparameters of the retrieval.

moderate degree of $k$NN guiding training is essential since $k$NN can help the model attend to hard examples, but excessive attendance of $k$NN-train also can bring the noise.

$\lambda$ **varies:** From Figure 1(b), we observe that model achieves optimal results on a 16-shot MR dataset when $\lambda$ is set to be 0.2 while attaining the best results on MR in the zero-shot setting when $\lambda$ is set to be 0.7. We think the model may require more reference when there is no data for training.

$k$ **varies:** As shown in Figure 1(c), the model performance in the 16-shot MR dataset fluctuates very little. In contrast, the result in the zero-shot MR dataset continues to improve as $k$ increases until it converges when reaching a threshold ($k = 256$). It illustrates that the $k$-NN retrieval provides more evidence for reference in zero-setting.

## E    Discussion of Limitation

**Analysis of Efficiency**    We make the comparison between LM-BFF (man), LM-BFF (+demo) and RETROPROMPT in speed on the MR dataset for the 16-shot setting. We observe that the speed of RETROPROMPT and LM-BFF (+demo) are approximately 1.12 and 20 times slower than LM-BFF (man) on the 16-shot MR dataset. The slow inference of LM-BFF (+demo) is due to the fact that they sample from the top $r\%$ instances ($r = 50$) for each class to use as demonstrations and vastly increase the length of the input, thus, increasing computational complexity significantly. And the bottleneck of computational speed is general limitations of retrieval methods, and our method is no exception. We will leave the engineering optimization about retrieval speed in our future work.

**Analysis of memory usage**    Actually, our method adopt FAISS tools for retrieval. FAISS is an excellent open-source library for fast nearest neighbor retrieval in high-dimensional spaces, which supports searching only from RAM, which involves k-means clustering for improving memory usage efficiency. Memory usage is negligible in the few-shot settings and acceptable in the full-data settings. Our retrieval process is performed mainly on CPU, and we compare the utilization of CPU with and without retrieval in the SST-2 full setting as follows:

- The CPU utilization was 46.2% with the retrieval process and 2.5% without it (Our CPU is Intel(R) Xeon(R) Silver 4210R CPU @ 2.40GHz with 40 cores).
- In terms of memory usage, adding retrieval requires about 2.5G more memory than not. One way to reduce resource usage is to store the datastore on the disk in advance, then read and release it in the retrieval process.