# OpenReview forum: "Decoupling Knowledge from Memorization: Retrieval-augmented Prompt Learning"
_NeurIPS.cc/2022/Conference — NeurIPS 2022 Accept_

### Official Review · Reviewer_SEU9 · 2022-07-03

**Rating:** 5
**Confidence:** 4
**Soundness:** 3 good
**Presentation:** 3 good
**Contribution:** 3 good

**Summary:**

This paper proposes a new semi-parametric approach for few-shot learning, where the prediction is made based on the combination of softmax probability from the language model (LM) and probability from retrieval over a knowledge store (in this paper, training data, either unlabeled or labeled). The paper proposes a few model components to make this approach work – the LM trained to be a good retriever (along with asynchronous update of the index), neural demonstrations (using nearest neighbors of the query from the knowledge store), and a modified objective (called guiding training in the paper). Experiments are done in a few-shot setup (where a few labeled examples leads to a knowledge store) and in a zero-shot setup (where an unlabeled training data is a knowledge store, and there is no parameter updates), and show strong results on 9 different NLP datasets (classification and information extraction) over competitive zero- and few-shot baselines.


**Questions:**

* kNN-based models are known to be effective when the size of the datastore is large. In this context, it is not intuitive to me how the model benefits from the kNN approach in a few-shot setup, where the datastore only consists of 4 or 16 examples.
* Is my understanding correct that, in a zero-shot setup, since there is no training phrase, the only difference from kNN-LM in terms of the model is the use of neural demonstration? Based on Table 1, the proposed model has significantly more improvements in a zero-shot rather than a few-shot? Are all gains here coming from neural demonstrations?
* Questions about Table 4:
    * w/o kNN-test: in this case, since the data store is not actually used, are gains mostly coming from better LM?
    * w/o kNN-train: in this case, is it identical to kNN-LM except there are neural demonstrations? (I know the knowledge store still looks different, but am curious in terms of modeling.) And is my understanding correct that this is identical to the main model in a zero-shot setup?
* Equation (6): It is not entirely clear to me how this objective achieves the problem mentioned in the section, and if this objective is mathematically sound. (e.g. why does it have to be the form of the multiplication between two log probability values?)
* Zero-shot setting: It seems debatable whether the use of training data (even if labels are not used) can be called zero-shot, although I understand authors followed the setup from previous work. Maybe it’s good to add a footnote noting that this is not zero-shot in a stricter sense?

**Limitations:**

The paper does not include limitations of work. I suggest authors to indicate limitations of the proposed model (e.g., when the proposed model may not work) as well as broader risk in applying large LMs in downstream tasks.

**Strengths And Weaknesses:**

### Strengths

* The idea of applying retrieval over a knowledge store for language model prompting is of interest in the community and is a timely topic.
* This paper is one of the first that applies the kNN-LM approach for downstream tasks, as far as I know.
* Empirical results are strong, outperforming a range of competitive baselines.
* Experiments also include extensive ablations, showing the impact of each model component.


### Weaknesses

* The idea is strongly based on the kNN-LM model (Khandelwal et al. 2021). In fact, the overall idea of combining softmax probability and probability from retrieval is entirely taken from kNN-LM, and how the model works at inference time is identical with kNN-LM, if I understand correctly. Although this paper is cited, it is cited only briefly (only cited twice, in Section 3.4 and Section 5). It looks to me that the paper should extensively discuss kNN-LM and describe the differences. (To clarify, I do believe there are substantial differences. They are just not discussed in the paper.)
* The model seems named after RETRO (Borgeaud et al. 2021), but the paper does not cite nor discuss the RETRO paper. In fact, I do not think the connection between this paper and RETRO is very tight – they’re only loosely related in the sense that both do retrieval, but what they retrieve, where they retrieve from, how they incorporate retrieval to LM, and what problems they are solving are all different. It is indeed not very intuitive to me why the model is named after RETRO, given that it is significantly more related to kNN-LM.
* It is not entirely clear to me what exactly are giving improvements, e.g., how the model gains from retrieval when the data store is extremely small (e.g., with 4 examples), how the zero-shot model achieves significant gains even with no training (which I believe is the main contribution of the paper), how the model still outperforms most competitive baselines (in Table 1) even without kNN-test that does not actually incorporate retrieval at test time, etc. I discussed them in more detail in the “Questions/comments” section.

(I am giving a borderline score due to the weaknesses mentioned here, but am happy to increase the score if they are resolved during the author rebuttal period.)

Borgeaud et al. 2021: https://arxiv.org/abs/2112.04426

---

> ### Author Response · Authors · 2022-08-02
> **Response to Reviewer SEU9 (Part1)**
>
> Thank you for the detailed and constructive comments.
>
> **R1:** As far as I know, the practice of combining softmax probability and probability from kNN was not invented by kNN-LM. It originated from Grave et al.(2017a)[1], then kNN-LM[2] followed Grave et al.(2017a) to produce the final kNN-LM distribution, Khandelwal et al. 2021 [3] further explore methods to improve its efficiency along various dimensions for kNN-LM. Our idea of interpolating the nearest neighbor distribution P<sub>kNN</sub> with the model distribution P<sub>LM</sub> using a tuned parameter λ **at test time** is indeed motivated by [1,2,3]. Our paper's overall motivation is to decouple knowledge from memorization with the comprehensive retrieval mechanism for prompt learning rather than the simple interpolation at test stages. We discuss the differences as follows:
>
> (a) Unlike kNN-LM, which solves language modeling tasks with generative models, we mainly focus on NLU tasks with prompt learning.
>
> (b) kNN-LM constructs datastore with sliding generative corpus and tokens, while we explicitly construct knowledge-store with regarding prompt-based instance representation as keys and label words as values, which is an initial exploration for prompt learning.
>
> (c) kNN-LM only incorporates the interpolation of the kNN distribution in the test phase, which offers a little gain in prompt learning (refer to the results of “w/o kNN-test” in Table 4). Therefore, we design the module of “kNN-train ” and “Neural Demonstration” to improve generalization ability during the input and training stages.
>
> Thanks a lot for your constructive suggestions and sorry for the incomplete discussion of the difference between ours and kNN-LM. We have added more comparisons and citations in Sections 3.1 and 3.4 of the revised draft.
>
> [1] Unbounded cache model for online language modeling with open vocabulary. NIPS 2017.
> [2] Generalization through Memorization: Nearest Neighbor Language Models. ICLR 2020.
> [3] Efficient nearest neighbor language models. EMNLP 2021.
>
> **R2:** Sorry for the missing reference (RETRO Borgeaud et al. 2021) and we have added it in Section 5 of our paper. We just name our model RetroPrompt because the generalization performance of prompt learning is improved with the retrieval mechanism across a comprehensive manner.

---

> > ### Author Response · Authors · 2022-08-02
> > **Response to Reviewer SEU9 (Part2)**
> >
> > Thank you for the detailed and constructive comments.
> >
> > **R3:**
> >
> > **For Q1:** 4-shot and 16-shot setup refer to 4 and 16 examples for each class. For example, the datastore of SST-2 datatset consists of 8 examples in the 4-shot setting. Our work aims to retrieve from the datastore to decouple knowledge from memorization. Despite only 4 examples in each class, the model benefits from the kNN approach as described below:
> >
> > (a) we can still retrieve related examples for each class as the demonstration, which can help the LM to learn by analogy and alleviate rote memorization.
> >
> > (b) we can still achieve the probability of kNN distribution as the prior external knowledge to guide the PLMs’ parameters attending to hard examples during the training process, which can calibrate the memory of a language model.
> >
> > (c) we can still incorporate kNN’s classification results into the final prediction at test time, which empowers the model to open book exams.
> >
> > **For Q2:** That’s a great question; there is no training phrase for OURS and baselines (except LOTClass) in a zero-shot setup. The difference from kNN-LM is not only the use of neural demonstration but also the construction of the datastore. The gains in the zero-shot setting come from the neural demonstrations, prompt-based construction of the datastore and interpolating the kNN’s results at inference time.
> >
> > **For Q3:**
> >
> > - **w/o kNN-test:** in this case, the datastore is also used for retrieving neural demo for demonstration learning and nearest neighbor for kNN guided training. The gains mostly come from better trained LM.
> > - **w/o kNN-train:** in this case, this is similar to kNN-LM except there are neural demonstrations from datastore (our datastore is indeed different and can be asynchronously refreshed). This setting is also similar to the main model in a zero-shot setup. The difference is that the model in a zero-shot setup doesn't involve training but “w/o kNN-train” involves training with few-shot data.
> >
> > **For Q4:** Sorry for the unclear parts. Capital $P_{k\text{NN}}$  is the probability mass obtained from kNN, the lower-case $p_{k\text{NN}}$  is the probability value of the gold class in the $P_{k\text{NN}}$ . The value of $p_{k\text{NN}}$  can reflect the difficulty of the instance to some extent, so we use it to re-weight the cross-entropy loss. We have stated this part more clearly in the revised draft.
> >
> > **For Q5:** Thank you for your excellent suggestion. We have added a footnote noting that this is not a strict zero-shot sense in the revised draft. Here we follow KPT that retrieves related knowledge without tuning parameters for zero-shot settings to compare fairly. The difference is that KPT retrieves from Related Words([https://relatedwords.org](https://relatedwords.org/)) to empower the verbalizer while we retrieve from unlabeled corpora to take advantage of our retrieval mechanism to improve the generalization of prompt learning.

---

> ### Author Response · Authors · 2022-08-10
> **Checking back**
>
> Dear Reviewer,
>
> We hope that you've had a chance to read our response. We would really appreciate a reply as to whether our response and clarifications have addressed the issues raised in your review, or whether there is anything else we can address.

---

### Official Review · Reviewer_YLj1 · 2022-07-08

**Rating:** 6
**Confidence:** 3
**Soundness:** 3 good
**Presentation:** 3 good
**Contribution:** 3 good

**Summary:**

This paper proposes an idea to retrieve from few-shot examples to prevent the prompt tuning process to bias toward learning from atypical examples. The combination of three techniques - knn-guided training, neural demonstration, and the knn-guided test-time prediction leads to a strong empirical performance across a broad set of tasks. Analysis shows that the proposed method effectively mitigates memorization.

**Questions:**

- For neural demonstration, I assume that the concatenation happens on the dimension of tokens? I suggest stating it more clearly in the paper.
- For section 4.4, did you just use 16 examples from the source dataset and not fine-tune on the target dataset at all?


**Limitations:**

Yes, the authors discuss about efficiency overhead of the proposed method in appendix.

**Strengths And Weaknesses:**

### Strengths
- The proposed method shows very strong empirical results compared to strong baselines
- It shows that even in the few-shot setting, retrieval-based approaches would be helpful for stabilizing and improving results without introducing additional parameters, and it could lead to useful application in a broader context.

### Weaknesses
- Details are missing, especially about the baseline and it weakens the compelling results. 1) KPT is a method to expand the verbalizers, what verbalizers did you get? I don't understand why the results would be worse than LM-BFF as it seems to be LM-BFF + additional verbalizers? 2) For LM-BFF, how did you get the scores with demonstrations in a zero-shot setting?
- Some design choices are not well justified or ablated: 1) Why specifically use FocalLoss? How does it compare to a linear combination of the original cross-entropy loss and a KNN loss similar to KNN-LM and it would be closer to the retrieval setup during test time? 2) Why does neural demonstration happen at the embedding layer?

---

> ### Author Response · Authors · 2022-08-02
> **Response to Reviewer YLj1**
>
> Thank you for the detailed and constructive comments.
>
> **R1:**
>
> 1). We reproduce the public codes of KPT for experiments on the datasets we chosen and select the same templates as ours. We also follow their paper to adopt Related Words ([https://relatedwords.org](https://relatedwords.org/)), a knowledge graph aggregated from multiple resources, including word embeddings, ConceptNet, WordNet, etc., as the external KB to expand the verbalizers. Here we absolutely did not weaken the compelling results. The reality is that the results of KPT are worse than LM-BFF (mainly for part datasets of GLUE) in the 16-shot setting but averagely better than LM-BFF in the 4-shot setting. Maybe KPT extends the label with the many additional related words and also introduces much noise.
>
> 2). In our zero-shot setting, the pseudo-labeled training sets are available to both RetroPrompt and LM-BFF(+demo). Thus LM-BFF(+demo) retrieves from that with examples that are semantically close to the input instance as the discrete demonstration.
>
> **R2**:
>
> 1). Sorry for the unclear parts. We do not use FocalLoss. We propose to leverage the kNN’s classification results as the prior external knowledge to guide the PLMs’ parameters attending to hard examples during the training process. FocalLoss is similar to  our motivation for calibration training and we just introduce FocalLoss for better understanding. On the other hand, our motivation of calibration training is different from a linear combination of the original cross-entropy loss and a KNN loss at training time. Moreover, kNN is a non-parameterized classifier, which is non-derivable. Thus, we cannot compare with that. We have stated this part more clearly in the revised draft.
>
> 2). The primitive demonstration is concatenated with the input example at the word embedding layer. Here we follow the previous practice to set neural demonstration at the embedding layer to participate in the information exchange of the current input instance.
>
> **R3:**
>
> - **For Q1:** Yes, for neural demonstration, the concatenation of input instance and neural demonstration happens on the dimension of tokens. Thanks for your excellent suggestion; we have stated it more clearly in Section 3.2 of the revised draft.
> - **For Q2:** Yes, in section 4.4, we prompt-tune the LM with 16 examples from the source dataset and not prompt-tune on the target dataset at all.

---

### Official Review · Reviewer_HTfi · 2022-07-11

**Rating:** 7
**Confidence:** 3
**Soundness:** 3 good
**Presentation:** 3 good
**Contribution:** 3 good

**Summary:**

They propose a method for better zero-shot and few-shot prompt learning called RETROPROMPT. To decoupling knowledge from memorization, they introduce knowledge-store which contains key-value as a [masked] label representation and its corresponding label word for the target data set. Using kNN on this neural knowledge-store, guided training and open-book inference can be implemented. By achieving best performances on 9 NLU tasks, the proposed method is shown to be effective.

**Questions:**

For the full fine-tuning, does the proposed model suffer from tremendous memory usage?

**Limitations:**

- Knowledge-store might be managed for full fine-tuning on large training dataset
- As metioned in the conclusion, this paper only deals with NLU tasks with encoder models.

**Strengths And Weaknesses:**

[Strengths]
- Well written.
- The paper propose technically sound and smart idea for balancing between generalization and memorization.
- Various experiements prove the effectiveness of the proposed RETROPROMPT for zero-/few-shot learning.

[Weaknesses]
- Experiments on full fine-tuning for all 9 datasets would be better for understanding the proposed work (only 3 results out of 9 datasets are included in the paper).
- The proposed model may suffer from enormous memory usage, especially for the full fine-tuning.

---

> ### Author Response · Authors · 2022-08-02
> **Response to Reviewer HTfi**
>
> Thank you for the detailed and constructive comments.
>
> **R1:** Prompt learning has arisen to improve the few-shot learning of LMs significantly. Thus, we mainly focus on few-shot, zero-shot and cross-domain settings in the experiments. Due to text space limitations, we have complemented the performance of full-data prompt-tuning in Appendix F of the revised draft.
>
> **R2:** Actually, our method adopts FAISS tools for retrieval. FAISS is an excellent open-source library for fast nearest neighbor retrieval in high-dimensional spaces, which supports searching only from RAM, which involves k-means clustering for improving memory usage efficiency. Memory usage is negligible in the few-shot settings and acceptable in the full-data settings. Our retrieval process is performed mainly on the CPU, and we compare the utilization of the CPU with and without retrieval in the SST-2 full-data setting.
>
> - The CPU utilization was 46.2% with the retrieval process and 2.5% without it (Our CPU is Intel(R) Xeon(R) Silver 4210R CPU @ 2.40GHz with 40 cores).
> - In terms of memory usage, adding retrieval requires about 2.5G more memory than not. One way to reduce resource usage is to store the datastore on the disk in advance, then read and release it in the retrieval process.

---

### Official Review · Reviewer_puCA · 2022-07-11

**Rating:** 5
**Confidence:** 4
**Soundness:** 3 good
**Presentation:** 3 good
**Contribution:** 2 fair

**Summary:**

This paper introduces RetroPrompt, which constructs an open-book knowledge store from training instances and implements a retrieval mechanism during the process of input, training, and inference, thus equipping the model with the ability to retrieve related contexts from the training corpus as cues for enhancement.


**Questions:**

See the comments in weakness. Especially points 2 and 3.


**Limitations:**

Discussion of limitations is lacking.

**Strengths And Weaknesses:**

Strength:

1. The experiments are extensive and convincing.
2. The idea of using kNN as a grounding demonstration is technically sound and plausible.
3. The choices of datasets and tasks are diverse.

Weakness

1. The idea of using nearest neighbors to train (kNN LM), to inference (FiD), and the general idea of using retrieval modules on memories (Google’s REALM, and Meta’s RAG), have been proposed and well studied. It’s unclear to me what is the novelty of this work. It seems like a combination of different models in different stages. And none of the models mentioned above are included as baselines.

2. What’s the performance of your method on open-ended knowledge-intensive tasks. As the tasks chosen in this work are mainly short-text tasks that can be answered with closed-from short texts (mainly overlapped with LM-BFF). Other tasks such as MSMARCO and Wizard of Wkipedia normally require more complicated reasoning rather than simple keyword memorization.

3. It is important to include some information retrieval systems as baselines, since if you integrate the retrieval system into every stage of LM, what’s the benefit of using an LM for unreliable prediction? I would like to see some analysis in the comparison with some IR-based methods, such as DPR, BM25, etc (+ necessary post-processing modules), if applicable.

4. The writing can be improved, especially some informal expressions.

5. No discussion of limitations

---

> ### Author Response · Authors · 2022-08-02
> **Response to Reviewer puCA**
>
> Thank you for the detailed and constructive comments.
>
> **R1:**
>
> - The general idea of using retrieval modules on memories (Google’s REALM, and Meta’s RAG), has been proposed to retrieve from the **external** knowledge corpus (e.g., Wikipedia) with **pre-training** for a particular task (e.g., an open-domain question). While we develop RetroPrompt with the motivation of decoupling knowledge from memorization to help the model strike a balance between generalization and memorization (mainly for prompt-tuning) with retrieving examples from the **internal** training data, which is essentially different from the motivation of REALM and RAG to introduce external corpus for reasoning. In fact, it doesn't make sense to rigidly use REALM and RAG for pre-training based on internal training data. Thus, we can’t include REALM and RAG as baselines due to the different knowledge sources to retrieve.
> - Our method is more related to kNN-LM, which only uses **internal** corpus. KNN-LM and FiD both only use nearest neighbors for **the inference** **process. At the same time,** we aim to develop a comprehensive retrieval mechanism for **input, training (original design of “prompt-based datastore construction”,“kNN-train” and “Neural Demonstration”) and test process** to further improve **generalization** of prompt-tuning. Moreover, kNN-LM is proposed to solve language modeling tasks by retrieving sliding generative corpus at test time, while we mainly focus on prompt learning for NLU tasks. Thus, it is hard to compare with kNN-LM on the same tasks directly. But we will try to generalize our work to generative and knowledge-intensive tasks in the future.
>
> Note that we consider the comprehensive retrieval mechanism to improve the generalization of prompt learning rather than the simple combination of different models in different stages. We also adopt self-influence as our memorization
> scoring function to analyze the memorization process between fine-tuning, prompt learning and our RETROPROMPT. The final analysis results show that 1) the training instances with the highest memorization scores tend to be atypical, 2) RETROPROMPT generalize better than fine-tuning and convention prompt-tuning with decoupling knowledge from memorization to alleviate the rote of PLMs.
>
> **R2:**  As discussed in R1, our motivation is essentially different from retrieval-augmented generation for knowledge-intensive tasks, such as complicated reasoning. We focus on helping the model balance generalization and memorization without additional external knowledge, mainly for prompt-based NLU tasks. The tasks chosen in this work also involve long-text tasks (such as information extraction). We will try to extend our work for knowledge-intensive tasks in the future. and thanks again for your insightful suggestions.
>
> **R3:**  Our method mainly involves retrieving related examples of corresponding training data to further improve prompt-tuning generalization. As described in the paper (Line 165), we mainly adopt inner product similarity for retrieving nearest neighbors. Intuitively, RetroPrompt is orthogonal to previous IR-based approaches (such as DPR, BM25), which are aimed at searching relevance score between docs. We have conducted experiments to compare our representation-based similarity score with BM25 based in Section 4.6. The results in Table 5 representation-based similarity scores for kNN lead to better performance.
>
> **R4:**  Thanks for your suggestion; we have improved some informal expressions in the revised draft based on the constructive comments of all the reviewers.
>
> **R5:**  We are so sorry that we don't include a separate chapter discussing limitations due to text space limitations. Despite this, we have previously discussed the efficiency overhead of the proposed method in Appendix D. Thanks a lot for your constructive suggestions; we have further complemented the comprehensive discussion of limitations in Appendix D of the revised draft.

---

> > ### Comment · Reviewer_puCA · 2022-08-07
> > **Thank you for your response**
> >
> > Thank you for your response. Some of my concerns have been addressed, though I am not fully convinced about the response to weakness 1. In any case, I have increased my score.

---

> > > ### Author Response · Authors · 2022-08-09
> > > **Thank you for your commmets**
> > >
> > > Dear Reviewer puCA:
> > >
> > > Thank you again for your constructive comments and kindly response. We will carefully revise our paper. We have **compared RetroPrompt with several retrieval-related approaches in the above "Summary of Revisions"** for further understanding.

---

### Author Response · Authors · 2022-08-02
**Summary of Revisions**

Dear reviewers and AC,

We sincerely appreciate your valuable time and constructive comments.

We’ve uploaded a revised draft incorporating reviewer feedback. Modified text is shown in blue font. Below is a summary of the main changes:

- Add the comparison with kNN-LM in Sections 3.1 and 3.4.
- Section 3.2 states that the concatenation of neural demonstration and input instance happens on the dimension of tokens.
- Section 3.3 states that the connection between $P_{k\text{NN}}$  and the  $p_{k\text{NN}}$ .
- Add a footnote noting that this is not a strict zero-shot sense in Section 4.2.
- Add a comprehensive discussion of limitations in Appendix D, including the analysis of memory usage.
- Add more results of full-data prompt-tuning in Appendix F.

We briefly introduce the motivation, method, contribution, and comparison with several retrieval-related approaches as follows:

Motivation: decouple the knowledge from memorization by constructing an open-book knowledge-store from the training data; thus, referring to related knowledge could provide a strong enhancement signal to help the model strike a balance between generalization and memorization.

Method:

- Construct knowledge-store with regarding prompt-based instance representation as keys and label words as values, which is an initial exploration for prompt learning.
- Retrieve related examples for each class as the demonstration, which can help the LM to learn by analogy and alleviate rote memorization.
- Leverage the probability of kNN distribution as the prior external knowledge to guide the PLMs’ parameters attending to hard examples during the training process, which can calibrate the memory of a language model.
- Incorporate kNN’s classification results into the final prediction at test time, which empowers the model to open book exams.

Contribution:

- We propose to decouple knowledge from knowledge to balance memorization and generalization.
- The first retrieval-enhanced prompt-tuning approach that brings promising improvements to a wide range of NLU tasks in few-shot, zero-shot and cross-domain settings.
- The comprehensive exploration of leveraging the self-influence based memorization scoring function to analyze the memorization process between fine-tuning, prompt learning and our RetroPrompt.
- Code and datasets are in the supplementary material and will be released for reproducibility.

**Comparison of RetroPrompt with several retrieval-related approaches**:

|     Model     |   Retrieval Set    |     Main Tasks     |    Integration Process     |              Retrieved Content              |
| :-----------: | :----------------: | :----------------: | :------------------------: | :-----------------------------------------: |
|     RETRO     | external knowledge | language modeling |        pre-training        |                    chunk                    |
|     REALM     | external knowledge |   open-domain QA   |        pre-training        |                   passage                   |
|    kNN-LM     |  internal corpus   | language modeling |         only inference          |          (K,V)=(embedding, token)           |
| LM-BFF(+demo) |  internal corpus   |   Prompt for NLU   |           input            |                   passage                   |
|      KPT      | external knowledge |   Prompt for NLU   |         verbalizer         |         related words of the label          |
|  RetroPrompt  |  internal corpus   |   Prompt for NLU   | input, prompt-tuning, inference | (K,V)=(prompt-based embedding, label words) |

We hope our responses and revisions address all reviewers’ concerns.

We sincerely believe that these updates may help us better deliver the benefits of the proposed RetroPrompt to the NeurIPS community.

Thank you very much,

Authors.

---

### Author Response · Authors · 2022-08-09
**Follow-up response to all reviewers**

Dear reviewers, we sincerely appreciate any suggestions and welcome any more questions about our response.

All Authors.

---

### Meta-Review · Area_Chair_f1Tv · 2022-08-28

**Recommendation:** Accept
**Confidence:** Certain

**Metareview:**

The paper proposes RetroPrompt, which builds a knowledge-store with training examples and improves few-shot and zero-shot performance.

All reviewers appreciate the improvements over competitive baselines and the quality of presentation. The main weaknesses are the lack of ablations to clarify where the gains come from and unclear positioning with respect to previous works (KNN-LM, RETRO, REALM, RAG). The reviewers were unanimous in accepting and the authors addressed some of the issues raised.

**Award:**

No

---

### Decision · Program_Chairs · 2022-09-14

Accept